# Synergistic Effects of Layered Double Hydroxide and MoS$_2$ on the Performance of Lubricants

**Weidong Zhou [1], Yong Li [2], Shutian Cheng [3], Yongdi He [1], Jinou Song [1,*] and Qiang Zhang [2]**

[1] State Key Laboratory of Engines, Tianjin University, Tianjin 300350, China; wdzhou000@126.com (W.Z.); tju_phd_hyd@163.com (Y.H.)

[2] State Key Laboratory of Chemical Resource Engineering, Beijing University of Chemical Technology, Beijing 100029, China; liy002.lube@sinopec.com (Y.L.); 2021200960@buct.edu.cn (Q.Z.)

[3] SINOPEC Lubricant Co., Ltd., Beijing 100089, China; chengsht.lube@sinopec.com

\* Correspondence: songjinou@tju.edu.cn

**Abstract:** In this study, layered double hydroxide (LDH) and molybdenum disulfide (MoS$_2$) were used as additives to prepare lubricants. The morphology and particle distribution of the LDH and MoS$_2$ were characterized using a scanning electron microscope and a laser particle size analyzer, respectively. Thermogravimetric analysis was used to compare the performance of the lubricants at high temperature. The extreme pressure and wear resistance performance of the lubricants were tested using a four-ball machine and a fretting-wear machine. Then, the lubricants were applied in a bolt fastener. The loosening torque and surface wear condition at high temperature were compared. By adding LDH and MoS$_2$ to the lubricants, the extreme pressure and wear resistance performance and anti-seize performance at high temperature were improved significantly. The LDH showed better anti-seize performance than the MoS$_2$ because of its strong and stable structure at high temperature. The MoS$_2$ showed better anti-wear performance under a high load because of its soft layered structure. The MoS$_2$ with a larger particle size showed better extreme pressure performance under a high load. The LDH and MoS$_2$ played a synergistic effect under the conditions of high temperature and high load.

**Keywords:** layered double hydroxide; molybdenum disulfide; lubricants; particle distribution; anti-wear; bolt fastener; anti-seize





## 1. Introduction

Solid fillers, as an important component of lubricants, can enhance the load-bearing capacity and high-temperature performance, and they can prevent aging and deterioration [1–4]. Molybdenum disulfide (MoS$_2$), with its unique characteristics of friction reduction, wear resistance, high-temperature endurance, and anti-magnetic properties, is commonly used as an additive and widely applied in fields such as lubricants, semiconductors, and alloy manufacturing [5–12]. It is known that the structure and size of MoS$_2$ play a decisive role and therefore researchers are dedicated to developing techniques such as organic modification to control the structure and size, thereby obtaining better characteristics and broader applications [13–20]. Various morphologies of MoS$_2$, including fullerene structures, spherical, rod-like, tubular, and linear shapes with particle sizes ranging from approximately 10–10,000 nm, have been synthesized through methods like liquid-phase and solid-phase processes. Efforts are being made to develop MoS$_2$ technologies with different structures and sizes for use in industrial areas such as lubricants and semiconductors. Although nanoscale and various structured preparation techniques have been achieved in the laboratory, scaling up for industrial application remains a challenge and a primary direction. Small-sized MoS$_2$ can enhance the anti-wear performance of lubricants to a certain extent, but the negative effects due to agglomeration, detachment, and changes in the morphology of the frictional surfaces require further detailed investigation. Therefore,

continued research on the application of $MoS_2$ with different structures and sizes, especially its tribological performance as an additive, is necessary to achieve a broader range of industrial applications.

Layered double hydroxides (LDHs) are a class of supramolecular-structured, anionic, layered, clay-like composite materials, assembled through non-covalent interactions between positively charged main layer sheets and interlayer anions. Their general chemical formula is represented as $[M^{2+}_{1-x}M^{3+}_x(OH)_2]^{x+}(A^{n-})_{x/n} \cdot mH_2O$ [21,22], where $M^{2+}$ and $M^{3+}$ represent the divalent and trivalent metal cations that constitute the main layer sheets, with $Mg^{2+}$ and $Al^{3+}$ being the most common; the variable 'x' denotes the molar ratio of trivalent metal ions to the total of divalent and trivalent metal ions; $A^{n-}$ represents the interlayer inorganic or organic anions that balance the positive charge of the main layer sheets, with $CO_3^{2-}$ being the most common; and 'm' denotes the number of interlayer water molecules. Due to the unique layered structure of LDHs, the types and quantities of the main layer elements and interlayer guest ions are adjustable. The crystal structure and grain size are controllable. Through intercalation assembly, specific inorganic or organic species can be introduced, thereby significantly enhancing existing or obtaining new properties while achieving new structures [23–25]. LDHs can serve as halogen-free flame retardants [26,27], lead-free thermal stabilizers [28,29], insulation materials, ultraviolet gas barrier materials [30,31], and other green and environmentally friendly advanced composite materials [32,33]. They are applicable in fields such as building materials, petrochemicals, agriculture, and highway construction.

In this paper, LDH and $MoS_2$ are used as additives to prepare lubricants. The morphology and particle distribution of the LDH and $MoS_2$ are characterized using a scanning electron microscope and a laser particle size analyzer, respectively. Thermogravimetric analysis is used to compare the performance of the lubricants at high temperature. The extreme pressure and wear resistance performance of the lubricants are tested using a four-ball machine and a fretting-wear machine. Then, the lubricants are applied in a bolt fastener. The loosening torque and surface wear condition at high temperature are compared. This article innovatively applies LDH in lubricants, investigates the synergistic effect of hydrotalcite and molybdenum disulfide, and investigates the influence of the molybdenum disulfide particle size on the wear resistance and high-temperature performance of lubricants. High-temperature lubricants are usually used in harsh working conditions such as under a high temperature and high load. Through this research, theoretical support is provided for the application of LDH and $MoS_2$ in lubricants. It is very meaningful to study the mechanism of action and evaluation methods of lubricants under these conditions.

## 2. Experimental

### 2.1. Materials and Preparation

The following reagents were used without further purification: $Mg(NO_3)_2 \cdot 6H_2O$, $Al(NO_3)_3 \cdot 9H_2O$, urea, and anhydrous ethanol. All the experiments were conducted using deionized water. The grades and sources of the chemical substances are shown in Table 1.

**Table 1.** The grades and sources of the chemical substances.

| Chemical Substances | Chemical Grade | Source |
|---|---|---|
| $Mg(NO_3)_2 \cdot 6H_2O$ | >99.0%, (AR) | Aladdin (Shanghai, China) |
| $Al(NO_3)_3 \cdot 9H_2O$ | >99.0%, (AR) | Aladdin (Shanghai, China) |
| Urea | >99.5%, (AR) | Aladdin (Shanghai, China) |
| Anhydrous ethanol | >99.7%, (AR) | Fuyu Co., Ltd. (Tianjin, China) |
| Polyalphaolefins oil 6 (PAO6) | >99.0% | ExxonMobil |

MgAl LDH was synthesized using a urea-assisted hydrothermal method. Initially, $Mg(NO_3)_2 \cdot 6H_2O$ (0.2 mol), $Al(NO_3)_3 \cdot 9H_2O$ (0.1 mol), and urea (1 mol) were dissolved in 1000 mL of deionized water. The resulting mixed solution was then transferred to a stainless steel polytetrafluoroethylene container and subjected to a 24 h reaction at 120 °C.

The products were subsequently washed three times with water and once with ethanol. Afterwards, they were dried for 12 h, resulting in the formation of MgAl LDH.

Lithium complex, polyurea, complex calcium sulfonate, and bentonite were used as thickeners to prepare the experimental base greases, with synthetic hydrocarbon PAO6 as the base oil. These were named BG1, BG2, BG3, and BG4, respectively, with a penetration range of 26.50–29.50 mm and dropping points all above 300 °C (Table 2).

**Table 2.** The compositions and performance of the base grease samples.

| Sample | Thickener | Base Oil | Penetration, 0.1 mm | Dropping Point, °C |
|--------|-----------|----------|---------------------|--------------------|
| BG1 | Lithium Complex | PAO6 | 272 | 310 |
| BG2 | Polyurea | PAO6 | 275 | 304 |
| BG3 | Calcium sulfonate | PAO6 | 274 | 328 |
| BG4 | Bentonite | PAO6 | 273 | 338 |

The LDH and $MoS_2$ used for the experiments were provided by Beijing University of Chemical Technology. The different specifications of $MoS_2$ were named M1, M2, and M3. The $MoS_2$ was a commercial product, while the LDH was a laboratory-synthesized MgAl type. The LDH and $MoS_2$ were added to the base grease BG3 and dispersed using a three-roll mill to obtain different lubricant samples (Tables 3 and 4). Samples G1 and G3 contained 10% and 30% M1, respectively; samples G2 and G6 contained 10% and 30% LDH, respectively; sample G4 contained 20% M1 and 10% LDH; and sample G5 contained 15% M1 and 15% LDH. In samples G4-1 and G4-2, the contents of $MoS_2$ and LDH were the same as in sample G4, but with different types of $MoS_2$ used, namely M2 and M3, respectively.

**Table 3.** The compositions and performance of the lubricant samples with LDH and M1.

| Sample | Base Grease | M1 Content, % | LDH Content, % | Penetration, mm | Dropping Point, °C |
|--------|-------------|---------------|----------------|-----------------|--------------------|
| G1 | BG3 | 10 | 0 | 27.91 | 324 |
| G2 | BG3 | 0 | 10 | 27.06 | 331 |
| G3 | BG3 | 30 | 0 | 28.83 | 317 |
| G4 | BG3 | 20 | 10 | 28.32 | 320 |
| G5 | BG3 | 15 | 15 | 27.64 | 329 |
| G6 | BG3 | 0 | 30 | 27.59 | 335 |

**Table 4.** The compositions and performance of the lubricant samples with LDH and different $MoS_2$.

| Sample | Base Grease | $MoS_2$ Type | $MoS_2$ Content, % | LDH Content, % | Penetration, mm | Dropping Point, °C |
|--------|-------------|--------------|--------------------|----------------|-----------------|--------------------|
| G4-1 | BG3 | M2 | 20 | 10 | 28.45 | 319 |
| G4-2 | BG3 | M3 | 20 | 10 | 28.36 | 321 |

*2.2. Test Instrument and Methods*

(1) A scanning electron microscope (SEM) from ZEISS Corporation (Oberkohen, Germany) was used to observe the microstructures of the LDH and $MoS_2$. The crystal structure of the LDH was characterized by X-ray diffraction (XRD) with an instrument from Shimadzu Corporation (Kyoto, Japan) using Cu K$\alpha$ radiation ($\lambda$ = 0.15406 nm).

(2) The particle sizes of the LDH and $MoS_2$ were measured using a Laser Particle Size Analyzer BT-9300S from Dandong Bettersize Instruments Ltd. (Dandong, China), with water as the medium and a refractive index of 1.333, in accordance with ISO 13320 [34].

(3) The distributions of the LDH and $MoS_2$ in the lubricants were examined using a Trinocular Upright Metallurgical Microscope 55XA from Shanghai Optical Instrument Factory No. 6 (Shanghai, China).

(4) The penetration of the different lubricant samples was tested using a Grease Penetration Tester BF-38 from North Dalian Analytical Instrument Co., Ltd. (Dalian, China), with s sensitivity of 0.01 mm, following ASTM D217 [35].

(5) The dropping point of the different lubricant samples was tested using a Wide-Temperature Range Grease Dropping Point Tester SYP4111 from Weiyou Petroleum Instrument Manufacturing Co., Ltd. (Shanghai, China), following ASTM D2265 [36].

(6) A Thermal Gravimetric Analyzer (TGA) 2LF from METTLER-TOLEDO Measurement Technology Ltd. (Zurich, Switzerland) was used to measure the thermal weight loss of the different lubricants at high temperatures ranging from 500 to 800 °C, with a heating rate of 10–20 K/min, an argon flow rate of 50 mL/min, and a scale sensitivity of 0.1 μg, following ASTM E1868 [37].

(7) An Automatic High Load Extreme Pressure Friction Tester STD081 from Falex Corporation (Chicago, IL, USA) was used to test the sintering load PD value of the different grease samples, following ASTM D2596 [38], and the wear scars on the steel balls under different conditions, following ASTM D2266 [39].

(8) A High-Frequency Linear Vibration Rig SRV5 from Optimol Instruments Pruftechnik GmbH (Munich, Germany) was used to test the friction coefficient of the different lubricant samples, with a sensitivity of 0.001, referring to ASTM D5707 [40]. The steel ball used in the SRV testing was made of 52,100 bearing steel with a hardness of $60 \pm 2$ HRC, a diameter of 10 mm, and a roughness (Ra) of $0.025 \pm 0.005$ μm. The steel disc was made of 52,100 bearing steel with a hardness of $60 \pm 2$ HRC, a diameter of 24 mm, a height of 7.85 mm, and a roughness (Ra) of $0.040 \pm 0.005$ μm. The testing conditions were as follows: load of 400 N, temperature of 80 °C, amplitude of 1 mm, frequency of 10 Hz, and duration of 1 h.

(9) A Contour GT-K0 Optical Profilometer from Bruker GmbH (Saarbrucken, Germany) was used to measure the wear volume of the steel disc. The Vertical Scanning Interferometry (VSI) mode was used, with an adjacent pixel height difference greater than 135 nm and a maximum scanning length of 10 mm.

(10) A High-Temperature Anti-Seize Device from Tianjin University was used to test the anti-seize performance of the different lubricant samples at high temperature, referring to the standard MIL-PRF-907H [41]. The fasteners used were M10, with bolts made of B16 alloy steel according to ASTM A193 [42] and nuts made of 2H according to ASTM A194 [43].

## 3. Results and Discussion

### 3.1. Characterization of MoS$_2$ and LDH

As shown in Figure 1, the XRD pattern of the MgAl LDH displays a series of reflections at 2θ angles of 11.6°, 23.6°, 34.9°, 39.6°, 47.1°, 60.8°, and 62.2°, corresponding to the (003), (006), (012), (015), (018), (110), and (113) crystal planes, respectively. The crystallinity was calculated to be 81.47%, indicating that the synthesized MgAl LDH possesses a high degree of crystallinity.

Figure 2 depicts SEM images of the MoS$_2$. The image reveals irregular block-like structures characteristic of MoS$_2$ particles, resembling rocks with an internal layered structure similar to that of an onion. Additionally, the presence of small, detached flakes indicates a relatively soft texture. From Figure 2, it can be seen that the particle sizes of samples M1, M2, and M3 are approximately 10.0 μm, 4.0 μm and 1.0 μm, respectively. However, this is only the morphology of an individual particle. The MoS$_2$ utilized in this study is derived from natural mineral raw materials, accounting for its irregular shape and uneven size.

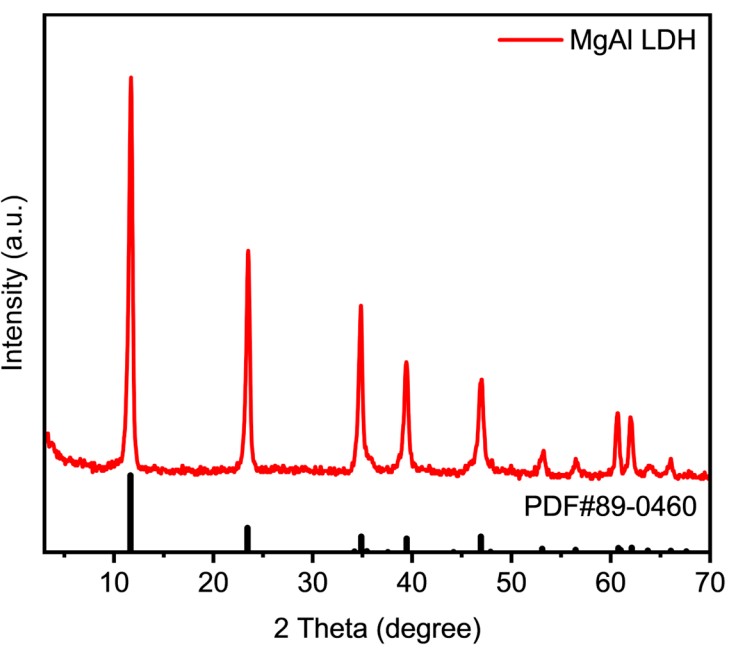

**Figure 1.** XRD pattern of the MgAl LDH.

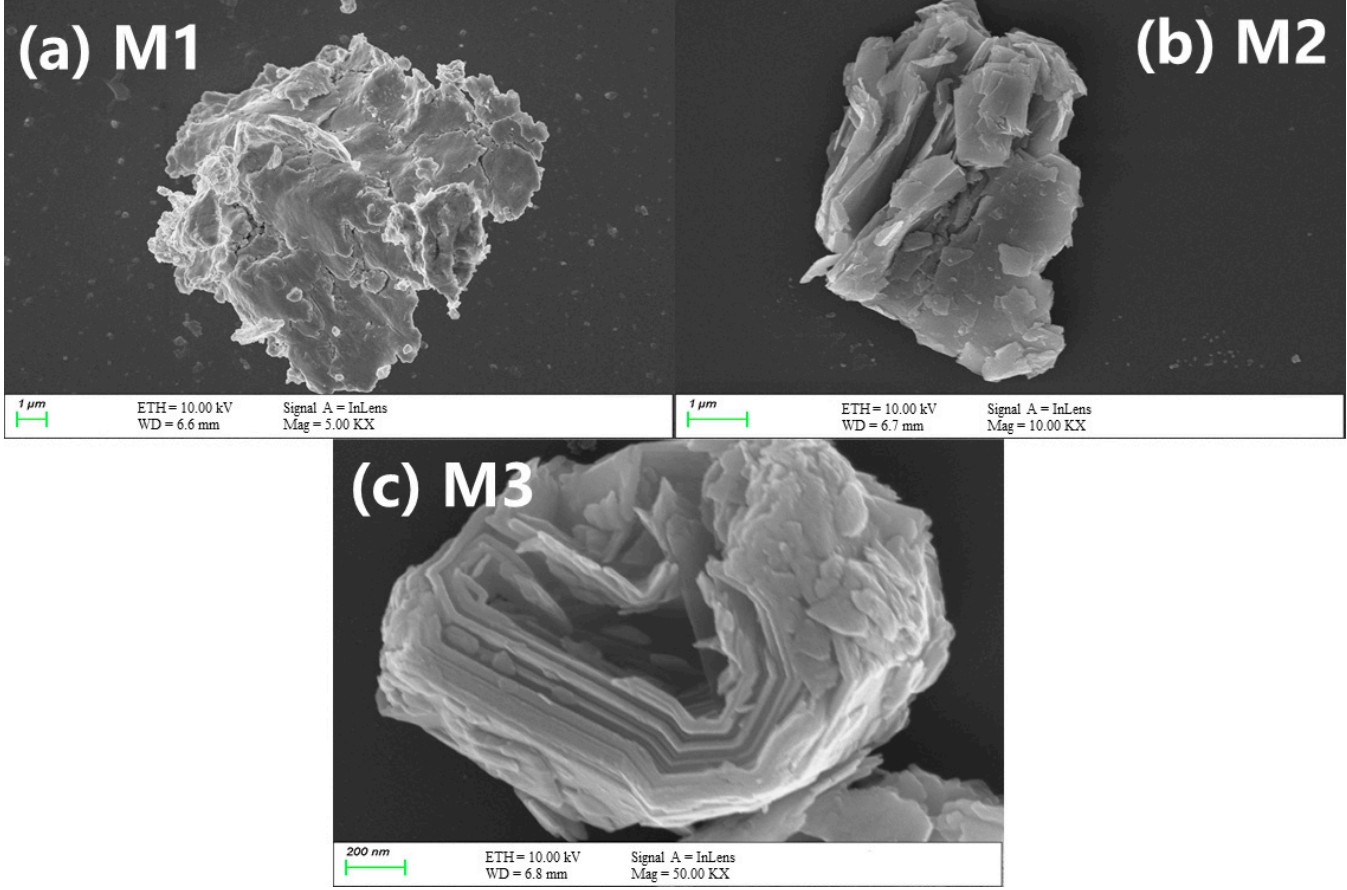

**Figure 2.** SEM images of the MoS$_2$ for the samples of M1 (**a**), M2 (**b**) and M3 (**c**).

Figure 3 presents a SEM image of the LDH. The image reveals a consistent hexagonal plate-like structure in the LDH, with the largest size reaching approximately 1 μm. The interior demonstrates uniformity, and the particle distribution appears to be exceedingly uniform, devoid of detached fragments, indicating a comparatively robust texture.

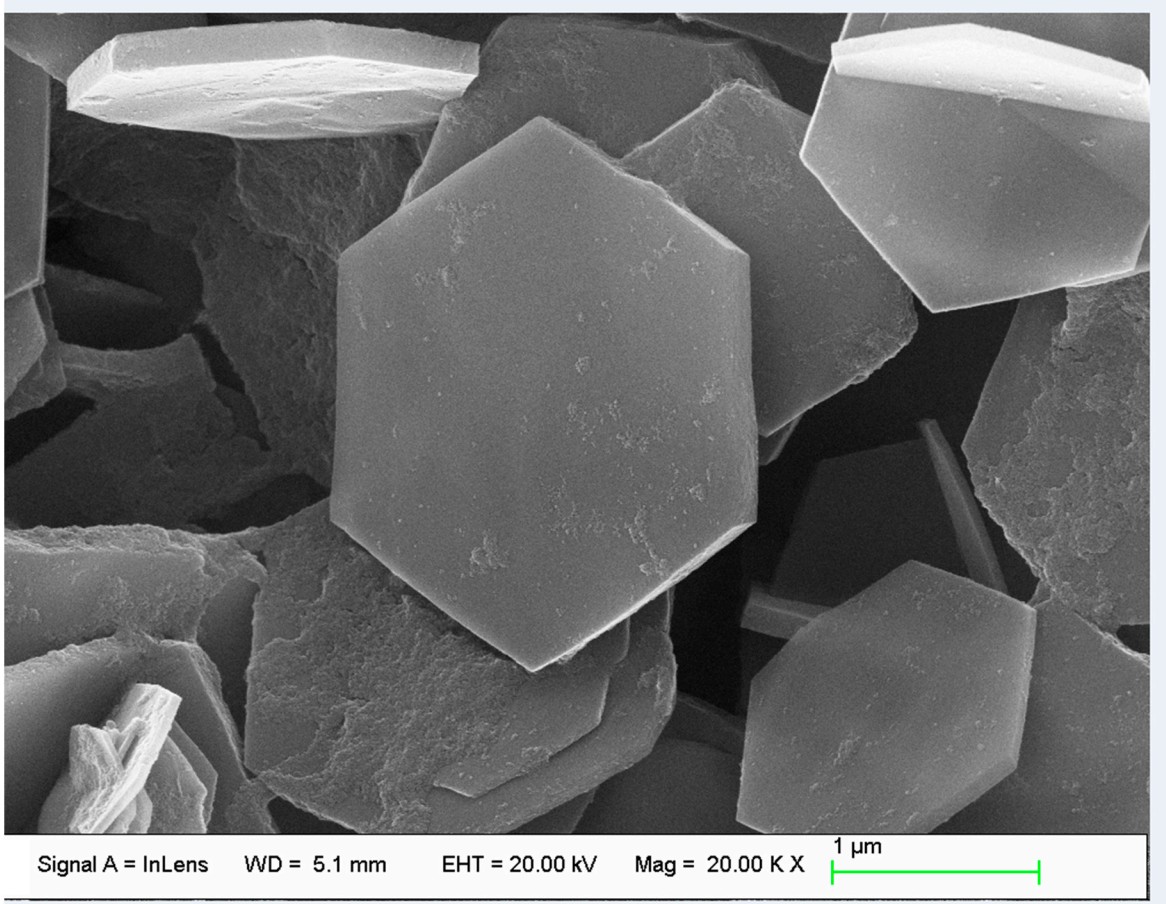

**Figure 3.** SEM image of the LDH.

The particle size distributions of the LDH and MoS$_2$ were analyzed using a laser particle size analyzer, and the results are presented in Figure 4 and Table 5. The analysis revealed that M1 demonstrates a particle size distribution ranging from 1.0 to 300.0 μm, with a D90 particle size of 81.510 μm. Similarly, M2 shows a particle size distribution ranging from 1.0 to 75.0 μm, with a D90 particle size of 25.580 μm, and M3 shows a particle size distribution ranging from 0.5 to 75.0 μm, with a D90 particle size of 11.580 μm. Successively, the particle sizes of M1, M2, and M3 decrease, leading to a gradual narrowing of the distribution. In contrast, the particle size distribution of the LDH ranges from 0.5 to 75.0 μm, with a D90 particle size of 3.073 μm, indicating that the LDH has smaller particle sizes and a more uniform distribution compared to the MoS$_2$.

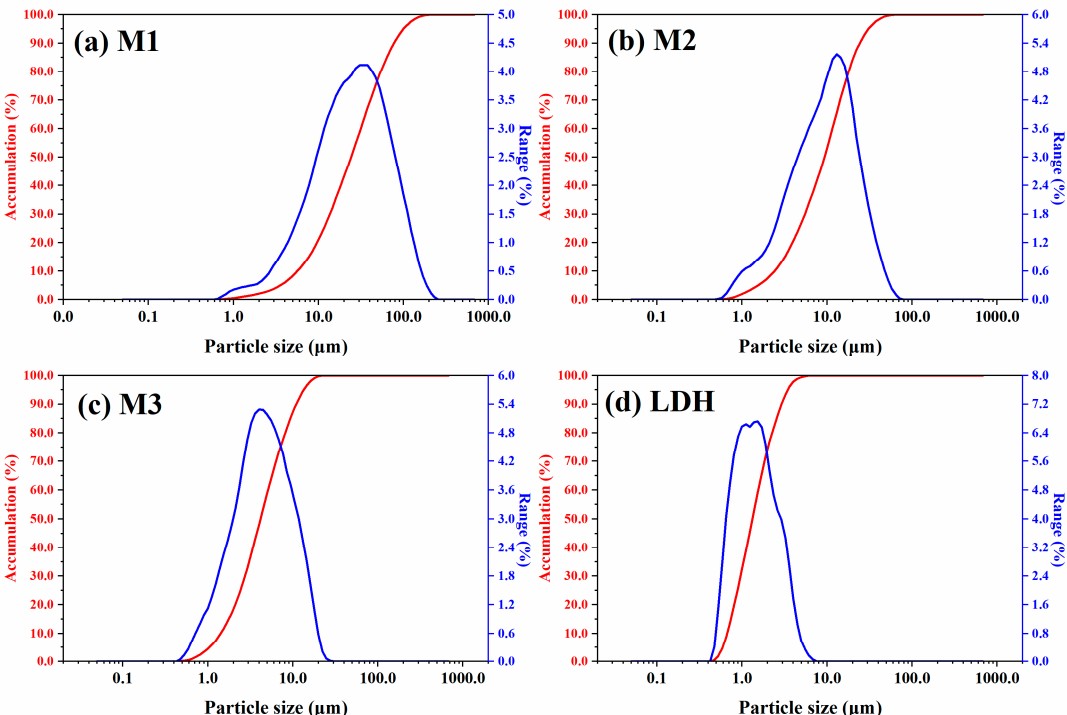

**Figure 4.** Particle size distributions of different MoS$_2$ samples by laser particle size analyzer (**a**–**d**) show the data for the samples of M1, M2, M3 and LDH, respectively.

**Table 5.** Particle sizes of different MoS$_2$ samples by laser particle size analyzer.

| Samples | Particle Size Distributions | | |
| --- | --- | --- | --- |
| | D50, μm | D90, μm | D100, μm |
| M1 | 25.430 | 81.510 | 239.400 |
| M2 | 9.828 | 25.580 | 71.930 |
| M3 | 4.381 | 11.580 | 26.000 |
| LDH | 1.398 | 3.073 | 7.060 |

### 3.2. Distributions of MoS$_2$ and LDH in Lubricants

Figure 5 presents microscopic images of the different lubricant samples magnified 200 times. In the base grease BG3, there are almost no visible solid particles present. However, after adding 30% M1, sample G3 exhibits evident solid particles with significant agglomeration, with the largest particle diameter reaching up to 50 μm. Upon adding 20% M1 and 10% LDH to sample G4, noticeable solid particles appear with less pronounced agglomeration, with the largest particle diameter approximately 30.0 μm. Adding 20% M2 and 10% LDH to sample G4-1 results in noticeable solid particles with minimal agglomeration, with the largest particle diameter around 20.0 μm, smaller than that in G4. Similarly, upon adding 20% MoS$_2$ M3 and 10% LDH to sample G4-2, conspicuous solid particles with a relatively uniform distribution are observed, with the largest particle diameter being less than 10.0 μm. According to the laser particle size analyzer, the D90 particle size of M3 is approximately 11.580 μm, larger than the LDH particle size, indicating that agglomeration is primarily attributed to the MoS$_2$. In summary, as the particle size of the MoS$_2$ decreases, the distribution of solid particles in the lubricant becomes more uniform and the agglomeration is less pronounced, contributing to the stability of the lubricants.

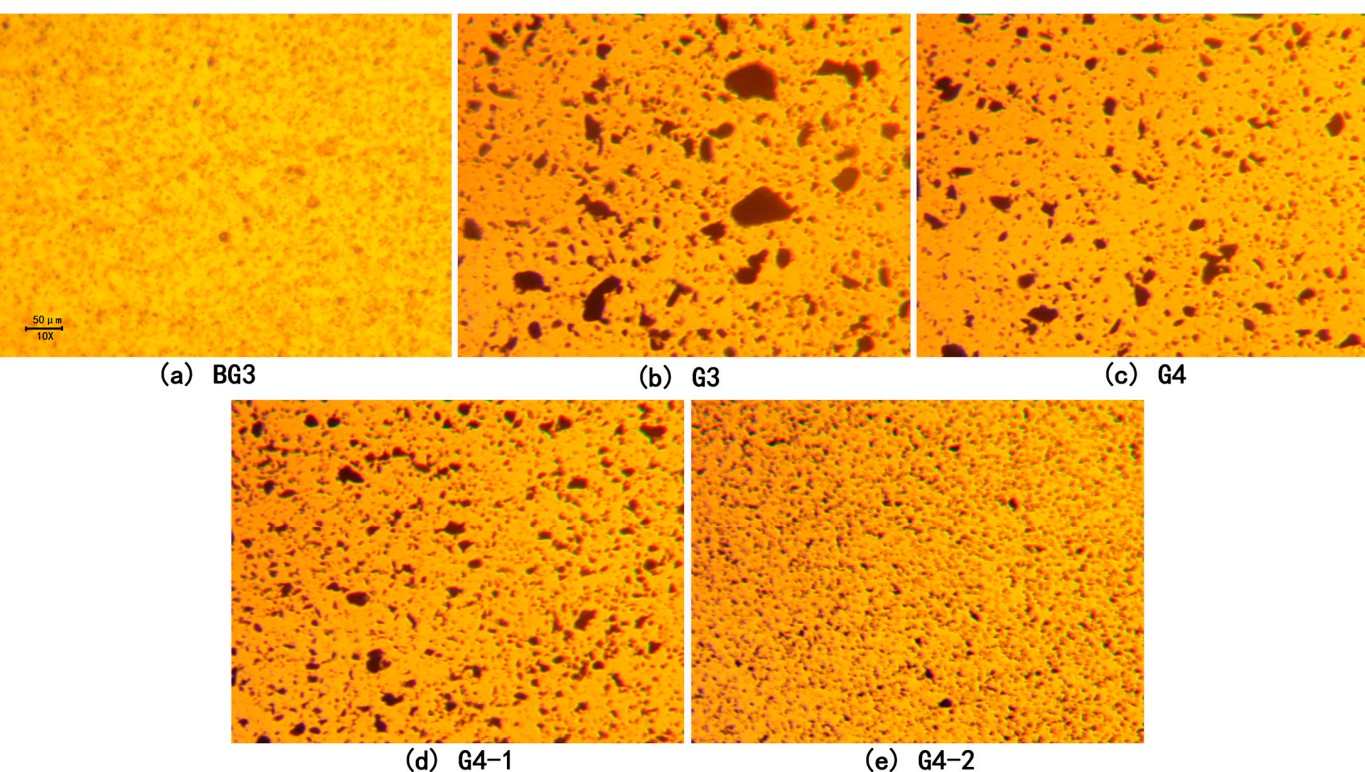

**Figure 5.** Microscopic images of different lubricant grease samples (with magnification 200×).

### 3.3. Thermogravimetric Analysis (TGA)

TGA was utilized to assess the weight loss of the different lubricants at elevated temperatures. The TGA results of the various base greases at 500 °C are depicted in Figure 6. At this temperature, the weight loss is 97.37% for BG1, 99.92% for BG2, 83.77% for BG3, and 92.15% for BG4. During the heating process, the mass of the base greases exhibits a sharp decline, stabilizing as the temperature approaches 500 °C. The base greases primarily consist of thickeners and PAO6 base oil, with the base oil typically starting to evaporate at temperatures exceeding 200 °C. As the temperature rises, the evaporation intensifies, potentially leading to decomposition into gas.

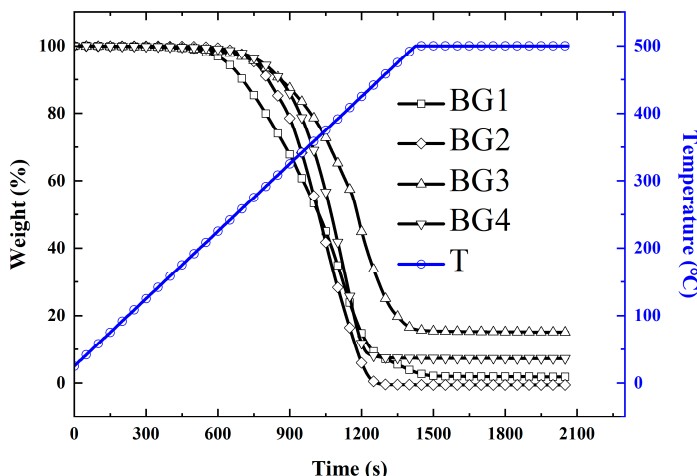

**Figure 6.** TGA test results of different base grease samples at 500 °C ('T' denotes temperature).

For BG1, with a lithium complex thickener, after exposure to 500 °C, the base oil and organic compounds in the lithium fatty acids evaporated or decomposed, primarily

leaving lithium oxide, constituting about 2.63% of the total mass, roughly corresponding to the lithium content of the base grease. BG2, with a polyurea thickener, showed almost complete evaporation or decomposition of the base oil and polyurea at 500 °C, leaving no residue. The thickener in BG3, complex calcium sulfonate, led to the partial evaporation or decomposition of the base oil and the thickener, ultimately leaving primarily calcium oxide, constituting approximately 17.23% of the total mass. In the case of BG4, with bentonite as its thickener, exposure to 500 °C caused the evaporation or decomposition of the base oil and thickener, resulting in primarily aluminum oxide and silicon dioxide, constituting about 7.85% of the total mass, closely corresponding to the content of the thickener in the base grease.

In summary, the base grease with complex calcium sulfonate shows the least weight loss after exposure to 500 °C, with the residue potentially acting as a solid lubricant at high temperatures. Therefore, for high-temperature applications, complex calcium sulfonate emerges as the preferable choice for the base grease.

The TGA results of the different lubricants at 800 °C are presented in Figure 7. It can be noted that the weight loss for base grease BG3 is 90.38% at 800 °C, an increase of 6.61% from the loss at 500 °C. The decomposition of organic matter in the sample intensified, with nearly all the residues being calcium oxide. After adding 10% $MoS_2$ M1, the weight loss for sample G1 is 81.58%, which is 8.8% lower than that of base grease BG3 and less than the theoretical value of 10%. This indicates that $MoS_2$ undergoes decomposition at high temperatures. Studies have found that $MoS_2$ starts to convert into molybdenum trioxide above 400 °C, leading to weight loss [44,45]. After adding 10% LDH, the weight loss of sample G2 is 80.39%, 9.99% lower than that of base grease BG3 and comparable to the theoretical value, suggesting that LDH remains stable and does not decompose at 800 °C. After adding 30% $MoS_2$ M1, the weight loss of sample G3 is 64.59%, 25.79% lower than that of base grease BG3 and less than the theoretical value of 30%. After adding 20% $MoS_2$ M1 and 10% LDH, the weight loss of sample G4 is 61.44%, 28.94% lower than that of base grease BG3 and 3% lower than G3. With the same solid content, the stable performance of LDH reduces the weight loss at high temperatures. In conclusion, under high-temperature conditions of 800 °C, $MoS_2$ decomposes, whereas LDH remains almost undecomposed, exhibiting superior high-temperature stability compared to $MoS_2$ and making it more suitable for high-temperature applications.

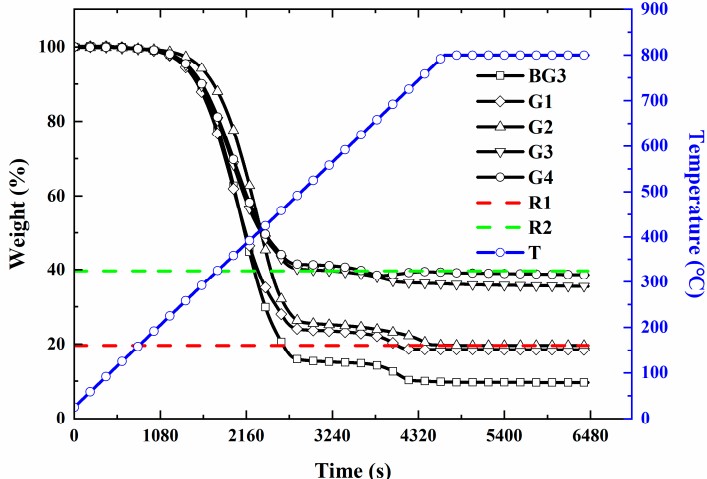

**Figure 7.** TGA test results of different lubricant grease samples at 800 °C ('T' denotes temperature, 'R1' denotes theoretical value of G1 and G2, 'R2' denotes theoretical value of G3 and G4).

### 3.4. Analysis of the Extreme Pressure and Anti-Wear Performance of Lubricants

The anti-load capacity of the lubricants was evaluated using the sintering load PD value from the four-ball tester, with the results presented in Figure 8. It is observed that the

PD value of base grease BG3 is 3089 N. After adding 10% M1, the PD value of sample G1 increases to 3923 N. Due to the active sulfur elements in MoS$_2$, a robust passivation film can form on the metal surface, enhancing the load-bearing capacity. After adding 10% LDH, the PD value of sample G2 is 3089 N, the same as base grease BG3. As the LDH content increases to 30%, the PD value of sample G6 remains at 3089 N, indicating that LDH lacks load-bearing capacity under high loads and does not chemically react with the metal surface to form a robust passivation film. After adding 30% M1, the PD value of sample G3 significantly increases to 6107 N compared to base grease BG3. Comparing samples G3, G4, and G5, it is evident that the PD value tends to rise with the increasing content of M1, playing a decisive role in the anti-load capacity of the lubricant. Comparing the PD values of samples G4, G4-1, and G4-2 reveals the different load-bearing capacities of M3, M1, and M2, leading to the reduced PD value of the lubricant. The smaller particle size of M3 is insufficient to cover the micro-contact surfaces, resulting in decreased load-bearing capacity. Therefore, the particle size of the MoS$_2$ has a certain impact on the load-bearing capacity of the lubricants.

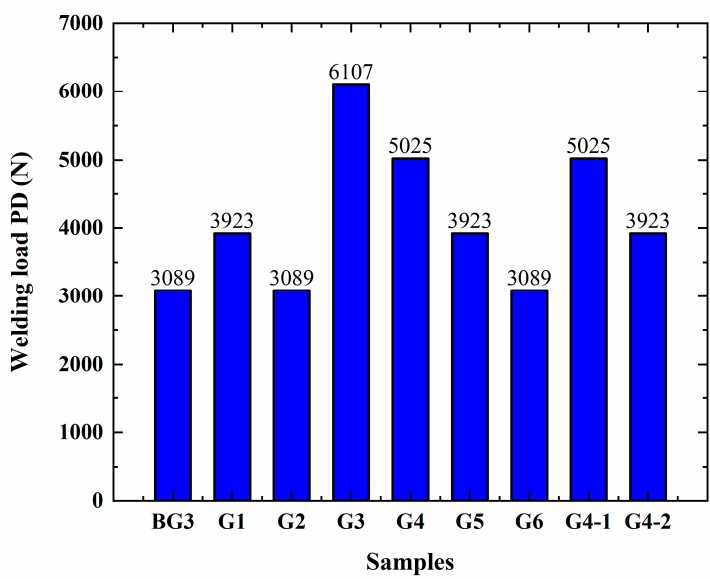

**Figure 8.** Weld-load PD value of different lubricant grease samples in the four-ball test.

The anti-wear capability of the lubricants was evaluated using the wear scar diameter from a four-ball tester, with the results presented in Figure 9. Under the conditions of a 400 N load, 1200 rpm speed, and 3600 s duration, the wear scar diameter of base grease BG3 is 0.550 mm. After adding 10% M1, the wear scar diameter of sample G1 decreased to 0.500 mm. After adding 10% LDH, the wear scar diameter of sample G2 is 0.475 mm, indicating that LDH exhibits superior anti-wear performance compared to M1 when added in the same quantity. As the contents of MoS$_2$ and LDH increase, the wear scar diameters of the lubricant samples show a decreasing trend, but the impact of MoS$_2$ is more significant than that of LDH.

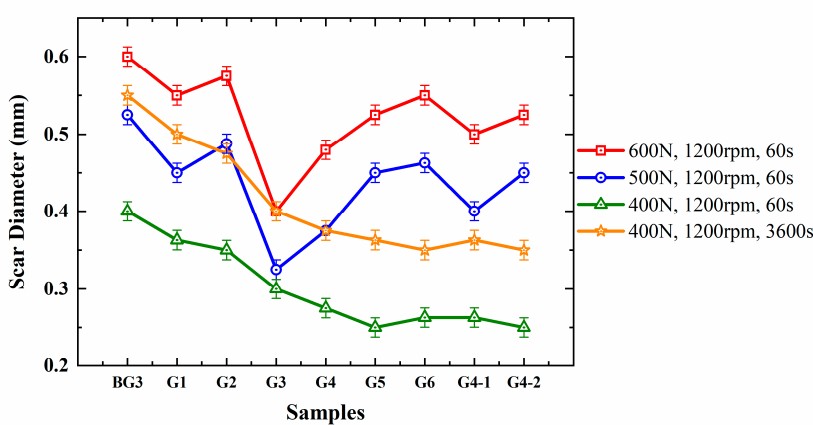

**Figure 9.** Scar diameters of different lubricant grease samples in the four-ball test.

Under low-load and short-duration conditions of 400 N load, 1200 rpm speed, and 60 s duration, the wear scar diameter of G2 is slightly smaller than that of G1, demonstrating that LDH performs better than M1 in terms of the anti-wear capability. However, as the load increases to 500 N and 600 N, the wear scar diameter of G2 become slightly larger than that of G1, indicating that the anti-wear capability of LDH is lower than that of M1 under an increased load, reflecting the lower load-bearing capacity of the LDH.

Comparing samples G4, G4-1, and G4-2 under conditions of 400 N load, 1200 rpm speed, and 3600 s duration, the wear scar diameter of G4-2 is 0.350 mm, smaller than those of G4 and G4-1. Due to the smaller particle size of M3 than those of M1 and M2, it can more easily reach the friction surface, exhibiting superior anti-wear performance. However, as the load increases to 500 N and 600 N, the wear scar diameter of G4-2 increases, indicating that under high-load conditions, the anti-wear capability of $MoS_2$ decreases with a reduction in the particle size. In summary, the load-bearing capacity of $MoS_2$ decreases with smaller particle sizes. Under low-load conditions, the smaller-sized $MoS_2$ exhibits better anti-wear performance. However, as the load increases, the larger-sized $MoS_2$ demonstrates superior anti-wear capabilities.

In summary, LDH demonstrates excellent anti-wear capabilities under low-load conditions, but its load-bearing capacity diminishes under high-load situations. $MoS_2$ exhibits outstanding anti-wear capabilities under high-load conditions, with its load resistance becoming more pronounced as the load increases. As the particle size of the $MoS_2$ increases, its anti-wear capability under low loads weakens, but its anti-wear capability under high loads strengthens, meaning that its load-bearing capacity is enhanced. The synergistic effects of $MoS_2$ and LDH can both increase the load-bearing capacity of the lubricant and enhance its anti-wear capability.

*3.5. Analysis of Fretting-Wear Performance of Lubricants*

The SRV test rig was utilized to evaluate the friction coefficients of the lubricants under fretting conditions, and an optical profilometer was used to assess the morphology and wear volume of the friction surfaces. The SRV friction coefficient test results of the different lubricants are shown in Figures 10 and 11. As illustrated in Figure 10, under the conditions of 400 N and 10 Hz, the friction coefficient of base grease BG3 is 0.136. After adding 10% LDH, the friction coefficient of sample G2 is 0.130, slightly lower than the base grease. After adding 10% M1, the friction coefficient of sample G1 decreases to 0.118, significantly lower than the base grease, indicating that M1 has superior fretting-wear resistance under high loads compared to LDH. After adding 30% LDH, the friction coefficient of sample G6 fluctuates between 0.094 and 0.116, showing more considerable variation. The increasing amount of LDH initially exhibits superior fretting-wear resistance, but the friction coefficient increases with the duration of the test. After adding 30% M1, the friction coefficient of sample G3 is 0.082, significantly lower than the base grease and markedly superior to other

samples. This illustrates that M1 has excellent fretting-wear resistance under high loads, attributed to its 'onion-like' layered structure. Comparing G4 and G5, it can be seen that as the amount of M1 increases, the sample's friction coefficient decreases.

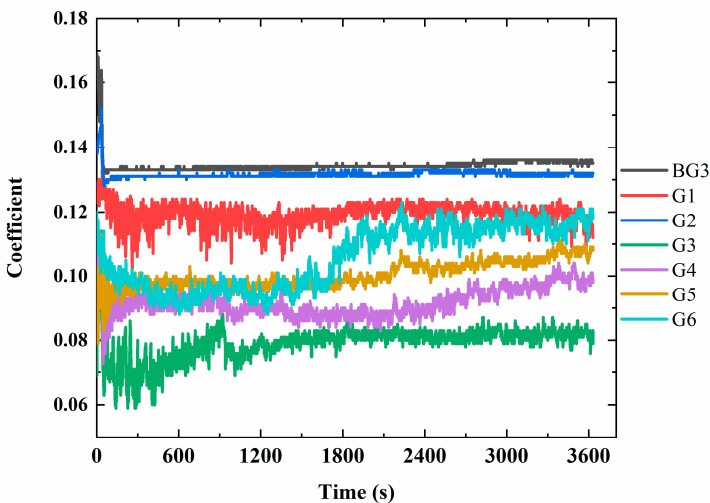

**Figure 10.** Coefficients of different lubricant grease samples at 400 N and 10 Hz in the SRV test.

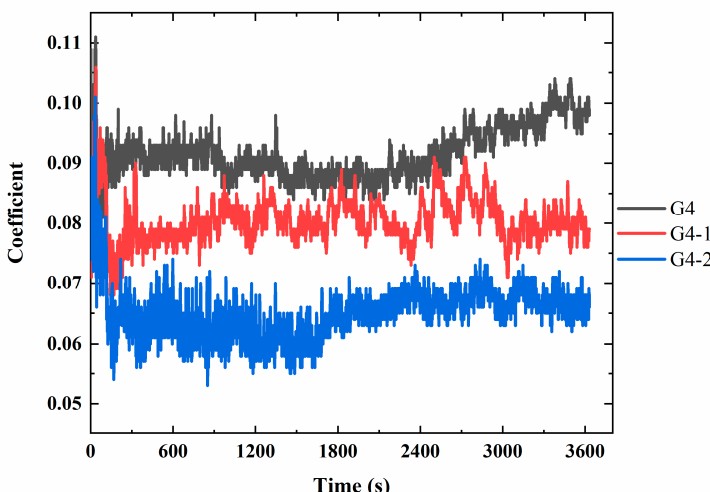

**Figure 11.** Coefficients of different lubricant grease samples with different MoS$_2$ at 400 N and 10 Hz in the SRV test.

As shown in Figure 11, under the conditions of 400 N and 10 Hz, the friction coefficient is 0.093 for sample G4, 0.080 for sample G4-1, and 0.066 for sample G4-2. With the same MoS$_2$ content, the particle size has a significant impact on the friction coefficient. As the particle size of the MoS$_2$ decreases, the friction coefficient also shows a decreasing trend.

After the SRV testing of the lubricant samples, the wear scars on the steel discs are magnified 50 times under a microscope and presented in Figure 12. It is observed that base grease BG3 shows noticeable scratches, revealing the metal substrate, indicating continuous rupture of the oil film under high-load reciprocating motion, resulting in a higher coefficient of friction. After adding 10% M1, the wear scar of sample G1 become darker, with shallower and more uniform scratches. A high local temperature during high-load reciprocating motion causes metal discoloration on the friction surface. After adding 10% LDH, the wear scar of sample G2 is even darker than G1, with some areas turning deep blue, indicating a higher local temperature, yet without obvious metallic scratches. After adding 30% M1, the wear scars of sample G3 are uniform, with no apparent metallic scratches, and the color

is black, showing the formation of a protective film on the metal surface. After adding 30% LDH, sample G6 shows no obvious scratches but is uneven, with some metallic areas, indicating that the protective film formed by LDH on the metal surface is neither robust nor uniform, leading to an increase in the coefficient of friction due to the continuous rupture of the protective film. Comparing G4 and G5, it is found that as the M1 content decreases, the scar color become lighter, but the scratches are more apparent. Comparing G4, G4-1, and G4-2, it is observed that as the particle size of the $MoS_2$ decreases, the wear scar color darkens, and the scratches become shallower, indicating that the smaller particle-sized $MoS_2$ forms a more evident protective film on the friction surface, resulting in superior anti-wear performance and a lower coefficient of friction.

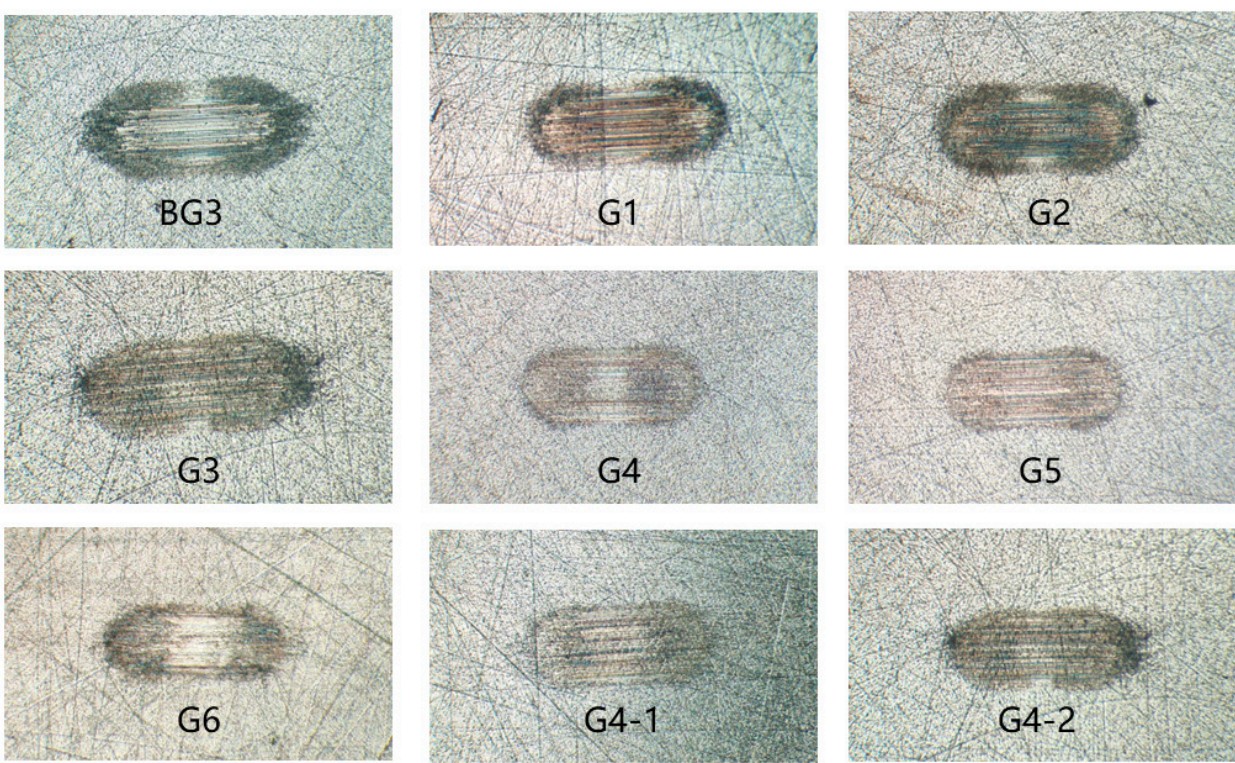

**Figure 12.** Columnar plate scars of different lubricant grease samples in the SRV test (with magnification 200×).

The wear volume of the SRV steel disc wear scars is measured using a Contour GT-K0 optical profilometer, and the results are shown in Figure 13. It is observed that the wear volume of base grease BG3 is 1,855,101 $\mu m^3$. After adding 10% M1, the wear volume of sample G1 decreases to 1,229,817 $\mu m^3$ (a reduction of 33.7% compared to base grease BG3). Increasing the M1 content to 30%, the wear volume of sample G3 is 534,510 $\mu m^3$ (a reduction of 71.2% compared to base grease BG3). After adding 10% LDH, the wear volume of sample G2 is 1,450,974 $\mu m^3$ (a reduction of 21.8% compared to base grease BG3). Increasing the LDH content to 30%, the wear volume of sample G6 is 980,699 $\mu m^3$ (a reduction of 47.1% compared to base grease BG3), much lower than that of M1. After adding 20% M1 and 10% LDH, the wear volume of sample G4 is 694,218 $\mu m^3$ (a reduction of 62.6% compared to base grease BG3), significantly better than G6. As the M1 content decreases and the LDH content increases, the wear volume of sample G5 is 870,828 $\mu m^3$, positioned between G4 and G3, and it is better than G6. It is evident that M1 has superior anti-wear performance compared to LDH, consistent with the trend of the change in extreme pressure performance of LDH. As the content increases, the anti-wear performance becomes more pronounced. Sample G4-2 has the smallest wear volume at 404,752 $\mu m^3$. Comparing G4, G4-1, and G4-2,

it is found that as the particle size of the $MoS_2$ decreases, the anti-wear capability of the samples increases.

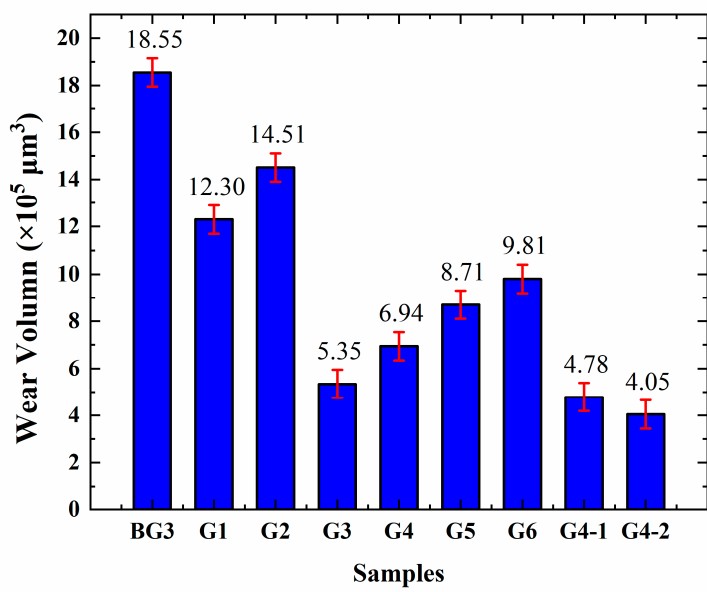

**Figure 13.** Wear volumes of different lubricant grease samples in the SRV test.

In summary, under high-load reciprocating motion conditions, M1 forms a robust and distinct protective film on the friction surface, leading to a reduction in the coefficient of friction, shallower scratches, and decreased wear volume. The protective film formed by LDH on the friction surface is less evident. As the wear duration extends, the film might rupture and wear out more intensively, resulting in an increased coefficient of friction, deeper scratches, and an increased wear volume. As the particle size of the $MoS_2$ decreases, the $MoS_2$ particles distribute more evenly within the lubricant and have a larger specific surface area, more readily accessible to the micro-friction surfaces, thus exhibiting superior fretting-wear resistance.

### 3.6. Analysis of High-Temperature Anti-Seize Performance of Lubricants

A high-temperature anti-seize device was used to test the anti-seize performance of different lubricant samples at high temperature. The lubricants were applied to five fasteners in the device with a pre-load torque of 160 N·m. The device was then placed in a muffle furnace and maintained at 800 °C for 6 h, followed by cooling to room temperature. The loosening torque of the five fasteners was tested, and the wear condition of the contact surface between the nuts and the panel was observed.

The results of the fastener-loosening torque test are presented in Figure 14. G0 represents the fastener without lubricant application, while the others indicate fasteners with the corresponding lubricants applied. It can be seen that after baking at 800 °C without lubricant, the average loosening torque of the fasteners is 202.6 N·m, indicating high torque and difficulty in disassembly. When base grease BG3 is applied, the average loosening torque of the fasteners reduces to 137.2 N·m. The TGA test reveals that mainly the calcium carbonate particles in the base grease play a lubricating role at high temperature. When sample G1 is applied, the average loosening torque of the fasteners is 119.8 N·m and 98.6 N·m for sample G2. After increasing the content of M1 and LDH, the average loosening torque of samples G3 and G6 further decreases, with G6 showing a larger reduction. It is evident that adding M1 and LDH to base grease BG3 reduces the average loosening torque of the fasteners and significantly improves the high-temperature resistance of the lubricants. LDH exhibits superior high-temperature performance compared to M1, consistent with the TGA test results. When both M1 and LDH are added, as in sample G4, the average loosening torque is 73.4 N·m. Comparing G4, G5, and G6, it is observed that as the relative content of

LDH increases, the average loosening torque of the fasteners decreases. Thus, under the high-temperature condition of 800 °C, LDH exhibits superior lubrication effects compared to M1.

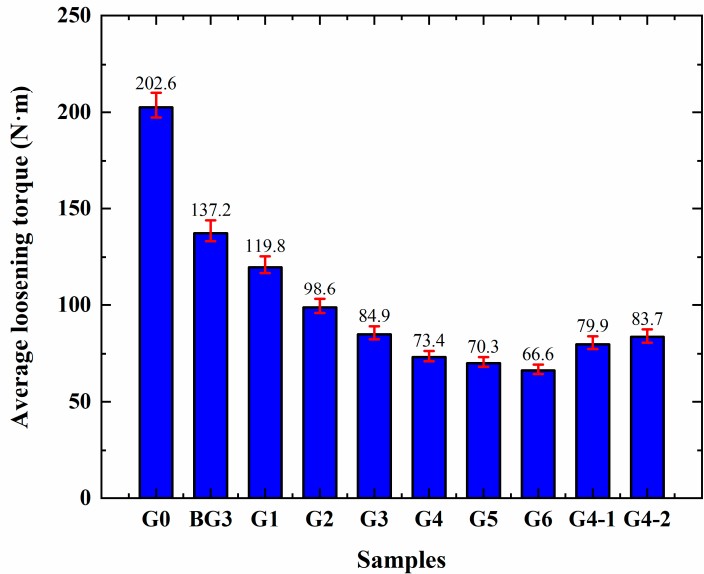

**Figure 14.** Average loosening torques of fasteners with different lubricants after heating at 800 °C and 6 h.

Comparing the data concerning G4, G4-1, and G4-2, it is observed that the lubricants exhibit different high-temperature resistance effects when $MoS_2$ with different specifications is added. As the particle size of the $MoS_2$ decreases, the loosening torque of the fasteners increases, and the greater the difference in the particle size, the larger the change in the loosening torque. This indicates that $MoS_2$ with a relatively larger particle size has superior high-temperature performance, presumably due to the larger $MoS_2$ particle having a more robust and stable structure at high temperature.

Images of the fastener nuts and contact surfaces are shown in Figure 15. As depicted in Figure 15a, after undergoing a pre-load torque of 160 N·m and baking at 800 °C, for the fastener G0 without lubricant, all five nuts and panel ends show severe wear, with completely scratched contact surfaces, resulting in excessive loosening torque and difficult disassembly. For the fasteners treated with base grease BG3, all five nuts and panel ends show noticeable wear, with three contact surfaces having deep scratches. For the fasteners treated with lubricant G1, three nuts and panel ends exhibit wear, one of which is severely worn. For the fasteners treated with lubricant G2, two nuts and ends show some wear, with both having deeper scratches. For the fasteners treated with lubricants G3, G4, and G5, all five nuts and panel ends show no wear. For the fastener treated with lubricant G6, one nut and panel end have noticeable scratches, with the rest showing no wear. Under high-load conditions, the SRV friction coefficient of lubricant G6 gradually increases over time, leading to significant lubricant depletion. Thus, under high-load and high-temperature conditions, the addition of only LDH results in the lowest loosening torque and exhibits the best high-temperature performance, although some fasteners show wear, indicating insufficient anti-wear performance. Combining the extreme pressure, anti-wear, and SRV tests, it is evident that under high-load conditions, the extreme pressure and anti-wear performance of LDH are inferior to those of $MoS_2$. When both $MoS_2$ and LDH are added, the lubricant demonstrates good high-temperature resistance and extreme pressure anti-wear performance, indicating a beneficial synergistic effect.

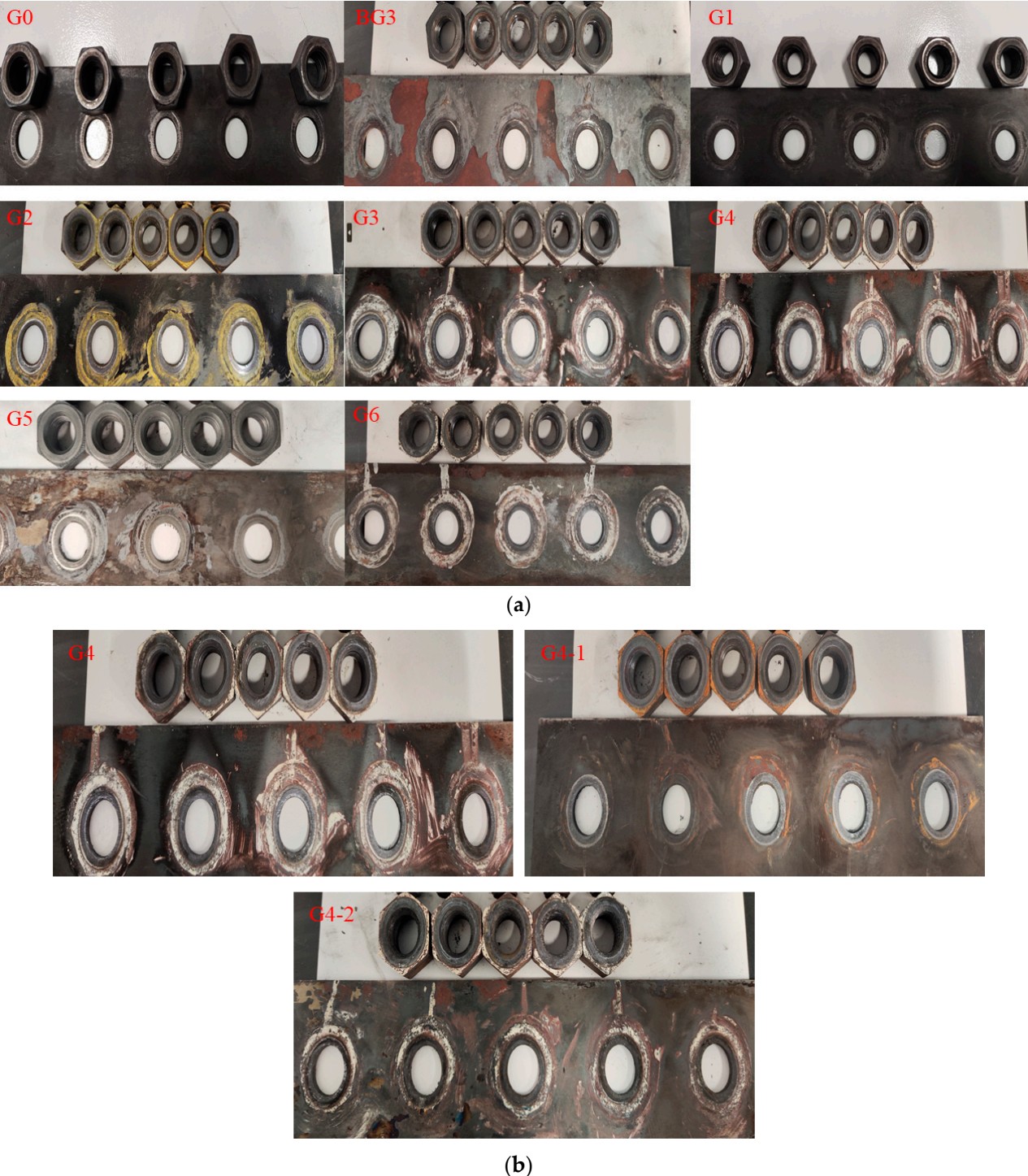

**Figure 15.** Images of fasteners with different lubricant grease samples after heating at 800 °C and 6 h: (**a**) G0/BG3/G1/G2/G3/G4/G5/G6; and (**b**) G4/G4-1/G4-2.

As shown in Figure 15b, the fasteners treated with lubricant G4-1 show no wear, while those treated with lubricant G4-2 have slight wear marks. Considering the extreme pressure and anti-wear performance. It can be seen that the smaller particle-sized $MoS_2$ has reduced load-bearing capacity, leading to the wear of the samples under high-load conditions. Therefore, as the particle size of the $MoS_2$ increases, both its high-temperature resistance and its load-bearing capacity are enhanced.

## 4. Conclusions

Based on the investigations carried out in this work, the conclusions are given as follows. By adding LDH and $MoS_2$ to lubricant, the extreme pressure and wear resistance performance and anti-seize performance at high temperature are improved significantly. The LDH shows better anti-seize performance than the $MoS_2$ because of its strong and stable structure at high temperature. The $MoS_2$ shows better anti-wear performance under high load because of its soft layered structure. The $MoS_2$ with larger particle sizes shows better extreme pressure performance under high load. The LDH and $MoS_2$ have a synergistic effect under the conditions of high temperature and high load. In terms of the wear resistance, sample G4-2 is the best. In terms of extreme pressure, sample G3 is the best. In terms of high temperature, sample G6 is the best. Based on this study, sample G4 has the best overall performance.

**Author Contributions:** Conceptualization, W.Z. and J.S.; methodology, W.Z., Y.L. and J.S.; validation, W.Z. and J.S.; formal analysis, W.Z. and J.S.; investigation, W.Z., Y.L., S.C. and J.S.; resources, J.S. and Q.Z.; data curation, S.C.; writing—original draft preparation, W.Z., Y.H. and J.S.; writing—review and editing, W.Z., Y.H. and J.S.; visualization, W.Z. and Y.H.; supervision, J.S.; project administration, W.Z. and J.S.; funding acquisition, W.Z. and J.S. All authors have read and agreed to the published version of the manuscript.

**Funding:** This study was supported by the National Natural Science Foundation of China (No. 52176122).

**Data Availability Statement:** The original contributions presented in the study are included in the article, further inquiries can be directed to the corresponding author.

**Conflicts of Interest:** Author Shutian Cheng was employed by the company SINOPEC Lubricant Co., Ltd. The remaining authors declare that the research was conducted in the absence of any commercial or financial relationships that could be construed as a potential conflict of interest.

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
