# Peer review of "Synergistic Effects of Layered Double Hydroxide and MoS2 on the Performance of Lubricants"

_lubricants, doi:10.3390/lubricants12050155_

Round 1

Reviewer 1 Report (Previous Reviewer 1)

Comments and Suggestions for Authors

The author response to all comments and now the manuscript is suitable to publish.  

Author Response

No suggestions were given by Reviewer 1.

Reviewer 2 Report (Previous Reviewer 2)

Comments and Suggestions for Authors

The quality of draft has been improved. However, some issues still need to be addressed.

1. SEM images should be revised to only keep important info like scale bar, no need to show 'ZEISS' and date/time.

2.Figure 11, XRD data should be revised.

3. Please summarize the main points in Conclusion part in sentences. Avoid using points by points

Author Response

Reviewer 3 Report (Previous Reviewer 3)

Comments and Suggestions for Authors

The authors addressed all my comments. I agree the publication of this manuscript. The PDF number of MgAl LDH should be mentioned.

Round 2

Reviewer 2 Report (Previous Reviewer 2)

Comments and Suggestions for Authors

N/A

Reviewer 3 Report (Previous Reviewer 3)

Comments and Suggestions for Authors

The authors addressed all my comments.

This manuscript is a resubmission of an earlier submission. The following is a list of the peer review reports and author responses from that submission.

Round 1

Reviewer 1 Report

Comments and Suggestions for Authors

Weidong, Z. et. al. investigated of LDH and MoS2 on the performance of lubricants. However, there are some points in the manuscript that need to be addressed. Here are comments;

1.     Line 10: “Layered double hydroxide” should be written as “layered double hydroxide”.

2.     Line 37: “10-10000nm” should be written as “10-10000 nm”. And please check space between number and unit entire the manuscript (Line 82, 158, 167, 169 and so on!).

3.     What is the novelty of this work?

4.     Line 77-81: Please check the subscription used in chemical formula. For example, “Mg(NO3)2” should be written as “Mg(NO3)2”.

5.     Please specified chemical grade and source for all chemicals.

6.     Please add the JCDPS reference pattern for XRD result. What is % crystallinity of MgAl LDH calculated from the XRD?

7.     What are morphology of M1, M2, and M3?

8.     Please recheck the size of LDH (Figure 3) again; the largest size is larger than 1 mm.

9.     The x-axis of TGA graph (Figure 6 and 7) must be temperature (°C).

10.  Please add error bars in Figure 8, 9 and 13

11.  The author should explain in the summary which sample is the best lubricant?

12.   Please recheck the format of reference. What is [J]?

Reviewer 2 Report

Comments and Suggestions for Authors

Layered double hydroxides (LDH), a typical two-dimensional material, consist of charged brucite-like hydroxide layers and exchangeable anions in the interlayer. This unique structural composition grants LDH nanoparticles excellent tribological properties as lubricant additives. However, the standalone lubricating capability of single LDH nanoparticles falls short in practical working conditions. Due to the abundance of positive charge on LDH nanoparticle surfaces, they readily absorb nanoparticles with a negative charge, such as MoS2, to form composite materials with superior lubricating performance.

In this paper, surface analysis techniques were utilized to examine the lubricating mechanism of the two types of LDH and MoS2. Meanwhile, the synergistic effect of the two additives is investigated. However, this paper shows poor novelty and the quality of the figures is unacceptable. Meanwhile, Section 4. Conclusions is too simple and needs further discussion. Hence, it is not recommended to be published in Lubricants

Comments on the Quality of English Language

  Section 2.2 Methods and Section 4. Conclusions should be changed to paragraphs rather than points.

Reviewer 3 Report

Comments and Suggestions for Authors

In this manuscript titled "Synergistic Effects of LDH and MoS2 on the Performances of Lubricants," the authors utilize LDH and MoS2 as additives to prepare lubricants. The results indicate that LDH shows better anti-seize performance, while MoS2 has a better anti-wear performance. They demonstrated a synergistic effect under conditions of high temperature and high load. In general, the experiments and data are sufficient to support their idea. However, this manuscript reads more like an experimental report than a paper. I encourage the authors to rearrange the manuscript. For instance, a) more motivation should be mentioned in the introduction, especially regarding the purpose of adding LDH. b) More transitions could be added between the results and discussion sections. Some detailed comments are attached below:

1. The authors are encouraged to include the standard peak of MgAl LDH in Figure 1 to provide a clearer indication.

2. A cleaner scale bar should be added in Figure 5.

3. On page 8, why does the G2 sample show a 10.75% improvement while the LDH was only 10%? Is this for the same reason that improves the stability of G4? Can the authors explain the reason for this? It is also a good idea to indicate the theoretical value in Figure 7 using a dashed line to make it clearer.

Reviewer 4 Report

Comments and Suggestions for Authors

Dear Authors,

The  article is well written and the study has delved in to the properties of lubricant oil in presence of LDH /MOS2 and both. From the study results it is conclusive that the hard nature of LDH is valuable for the thermal stability but the soft nature of MoS2 provide better wear resistance. I just have one qhstion about the fact that the use of higher LDH (30%) decrease the Wear volume of columnar plate in SRV test but no changes has been seen for the welding load but lessen in scar diameter. Is there any reason for such behaviour?